# Cost-effectiveness of routine adolescent vaccination with an M72/AS01$_E$-like tuberculosis vaccine in South Africa and India

Rebecca C. Harris [ID] [1,3,5✉], Matthew Quaife [ID] [1,5], Chathika Weerasuriya[1], Gabriela B. Gomez[2,4], Tom Sumner[1], Fiammetta Bozzani [ID] [2] & Richard G. White [ID] [1]

The M72/AS01$_E$ tuberculosis vaccine showed 50% (95%CI: 2–74%) efficacy in a phase 2B trial in preventing active pulmonary tuberculosis disease, but potential cost-effectiveness of adolescent immunisation is unknown. We estimated the impact and cost-effectiveness of six scenarios of routine adolescent M72/AS01$_E$-like vaccination in South Africa and India. All scenarios suggested an M72/AS01$_E$-like vaccine would be highly (94–100%) cost-effective in South Africa compared to a cost-effectiveness threshold of $2480/disability-adjusted life-year (DALY) averted. For India, a prevention of disease vaccine, effective irrespective of recipient's *M. tuberculosis* infection status at time of administration, was also highly likely (92–100%) cost-effective at a threshold of $264/DALY averted; however, a prevention of disease vaccine, effective only if the recipient was already infected, had 0–6% probability of cost-effectiveness. In both settings, vaccinating 50% of 18 year-olds was similarly cost-effective to vaccinating 80% of 15 year-olds, and more cost-effective than vaccinating 80% of 10 year-olds. Vaccine trials should include adolescents to ensure vaccines can be delivered to this efficient-to-target population.

[1] TB Modelling Group, TB Centre, and Centre for Mathematical Modelling of Infectious Diseases, Department of Infectious Disease Epidemiology, London School of Hygiene and Tropical Medicine, London, UK. [2] Department of Global Health and Development, London School of Hygiene and Tropical Medicine, London, UK. [3] Present address: Sanofi Pasteur, Singapore, Singapore. [4] Present address: Sanofi Pasteur, Lyon, France. [5] These authors contributed equally: Rebecca C. Harris, Matthew Quaife. ✉email: Rebecca.harris@lshtm.ac.uk

Over the last decade, tuberculosis has killed more people globally than any other single infectious pathogen[1]. One hundred years after its development, bacille Calmette–Guérin (BCG) remains the only licensed vaccine against tuberculosis. BCG primarily prevents extra-pulmonary tuberculosis in children[2], but the majority of the global burden of disease remains in adolescents and adults. New vaccines to prevent adolescent and adult tuberculosis are urgently needed. In 2018, the novel vaccine candidate M72/AS01$_E$ was shown 50% (95% CI: 2–74%) efficacious in preventing pulmonary tuberculosis disease in *Mycobacterium tuberculosis* (*M. tuberculosis*)-infected 18–50-year-olds[3], and may therefore be an effective supplement to BCG. More candidates are in development[4], while BCG revaccination in adolescence is also being explored[5].

Previous work has estimated the epidemiological impact of hypothetical or pipeline vaccines to prevent tuberculosis infection and/or disease[6–8], including the impact of a post-exposure vaccine on drug-resistant tuberculosis[9,10], and when delivered via routine immunisation of 9-year-olds accompanied by recurring mass campaigns to adolescents/adults[10,11]. Routine immunisation of only adolescents could prove more feasible and cost-effective than large mass campaigns, but the cost-effectiveness of such an approach has not been explored; in particular, it is critical to understand the age groups in which vaccination will be most cost-effective. This study estimates tuberculosis vaccine cost-effectiveness based on M72/AS01$_E$ phase 2B efficacy results to assess whether routine vaccination of adolescents could be cost-effective. Importantly it considers costs to both the health system and patients. Furthermore, it allows us to assess the relative importance of routinely vaccinating adolescents with disease-preventing vaccines that are (i) effective irrespective of whether the recipient is infected with *M. tuberculosis* at the time of vaccination ("pre- and post-infection efficacy"); or (ii) effective only when the recipient is already infected with *M. tuberculosis* ("post-infection efficacy"), see Box 1. This study is timely to guide investment in phase III and IV trials, alongside trial design and implementation decisions.

In the present study, we used a previously calibrated age-structured compartmental dynamic *M. tuberculosis* transmission model to assess the pop/ulation-level impact and cost-effectiveness of a new M72/AS01$_E$-like vaccine, delivered during 2025–2050 to adolescents in South Africa and India. We estimated impact on mortality and morbidity, alongside health service costs of routine vaccination and health service and patient costs averted from preventing tuberculosis disease. We assessed cost-effectiveness using country-specific cost-effectiveness thresholds. To inform programmatic planning, we simulate three scenarios of routine adolescent vaccination, assuming a coverage of 80% is reached among 10-year-olds and 15-year-olds, and that a lower coverage of 50% is reached among 18-year-olds, to reflect potential difficulties in reaching older adolescents in vaccination campaigns.

## Results

In both settings, all adolescent routine vaccination scenarios substantially reduced tuberculosis disease incidence rates in 2050, shown in Table 1. A vaccine with pre- and post-infection efficacy had a much greater impact on 2050 incidence rate than a vaccine with post-infection efficacy only. Depending on the vaccination scenario modelled, a pre- and post-infection vaccine had between a 9- and 23-fold greater impact on incidence rate than post-infection only in India, and between 5- and 13-fold greater impact in South Africa. Similarly, a vaccine with pre- and post-infection efficacy averted between 7 and 15-times more DALYs than a post-infection vaccine only in India, and between four and ten times more DALYs in South Africa. The greatest difference in impact between vaccines of different efficacy types was in the scenarios vaccinating 10-year-olds.

In South Africa, we estimate that any routine vaccination scenario, with either a pre- and post-infection or post-infection-only efficacious vaccine, would be highly cost-effective from the health system perspective (94–100% probability of being cost-effective across scenarios modelled), as shown in Table 1 and Fig. 1. Even the least cost-effective scenario (post-infection-only vaccine with 80% coverage among 10-year-olds) was much lower than the cost-effectiveness threshold ($2480) at $1241/DALY averted (95% CrI: $519.9, $2555) from the health system perspective. When patient costs were also included (societal perspective), the pre- and post-infection vaccine among 15- and 18-year-olds became cost-saving, or dominant, in all scenarios modelled (Table 1, Supplementary Fig. S8), though we note that this is compared to the cost-effectiveness threshold based on

---

**Box 1 ▌ Assumed vaccine mechanism of action, host infection status required for efficacy, and timing of administration relative to infection**

In this study, we only modelled new TB vaccines that reduced the occurrence of tuberculosis disease (prevention of disease or POD). None of the vaccines we modelled affected the probability of infection.

We also modelled vaccine types which were differentially effective depending on whether the recipient was infected by *Mycobacterium tuberculosis* (Mtb) (host infection status required for efficacy):

- *Pre- and post-infection efficacy (P&PI) vaccines.* These vaccines were effective *irrespective* of whether the recipient had previously been infected by Mtb or not, i.e., they were effective in those naïve to infection, those with latent Mtb infection, or those who had recovered from active disease. These vaccines conferred protection against tuberculosis disease by reducing the progression to disease following first-time infection or reinfection, or reactivation from latent infection, or relapse after recovery from active TB.
- *Post-infection efficacy (PSI) vaccines.* These vaccines were effective *only* when administered to individuals who were *already* infected with Mtb, i.e., either latently infected or recovered following active disease. The vaccine was *completely ineffective* if administered to an individual *naïve* to Mtb infection. Protection was conferred against disease due to reducing progression to disease following reinfection, or reactivation from latent infection, or relapse after recovery from active TB.
- *Pre-infection efficacy (PRI) vaccines.* These vaccines were not modelled in this study. They represent the scenario where a vaccine would be effective *only* when administered to an individual who is naïve to Mtb infection. They would be *completely ineffective* if administered to an individual with latent infection by Mtb, or who had recovered from active TB. Protection against disease would be conferred by reducing progression to disease after the first-time infection of an individual with Mtb.

We also assumed that there was no pre-immunisation testing for Mtb infection, so all individuals, at the specified coverage, in the target age group were administered the vaccine, regardless of infection status. Efficacy in a given recipient was determined solely by their underlying infection status and the characteristics of the vaccine.

**Table 1 Incidence rate reduction, incremental costs, DALYs averted, and incremental cost-effectiveness ratios for different vaccination programme scenarios compared to a counterfactual scenario without vaccination.**

| Scenario description | Incidence rate reduction in 2050 (%) | Incremental health system cost ($USD million) | DALYs averted | Cost per DALY averted—health system perspective ($USD) | Incremental health system and patient cost ($USD million) | Cost per DALY averted—societal perspective ($USD) |
|---|---|---|---|---|---|---|
| **South Africa** | | | | | | |
| *Pre- and post-infection efficacy* | | | | | | |
| 10-year-olds, 80% coverage | 11.3 (7.9, 14.3) | 22.7 (−24.3, 86.9) | 555,707 (424,020, 7,547,352) | 41.6 (−40.6, 163.5) | 9.8 (−40.8, 73.4) | 17.6 (−68.0, 137.5) |
| 15-year-olds, 80% coverage | 14.1 (9.2, 18.4) | 14.3 (−42.3, 85.3) | 665,283 (495,032, 966,186) | 20.6 (−56.2, 143.1) | −4.8 (−64.2, 65.0) | Dominant −7.2 (−86, 109.0) |
| 18-year-olds, 50% coverage | 9.5 (6.3, 12.4) | 11.0 (−26.3, 58.9) | 441,175 (332,273, 628,197) | 24.3 (−57.3, 140.8) | −1.6 (−44.2, 46.5) | Dominant −3.6 (−91.2, 113.8) |
| *Post-infection efficacy only* | | | | | | |
| 10-year-olds, 80% coverage | 0.9 (0.6, 1.3) | 70.6 (30.8, 133.7) | 55,141 (41,465, 79,327) | 1241.0 (519.9, 2555.0) | 68.9 (29.5, 132.4) | 1217.1 (502.5, 2526.1) |
| 15-year-olds, 80% coverage | 2.1 (1.5, 3.0) | 64.3 (23.9, 129.6) | 127,054 (91,551, 186,339) | 497.1 (172.2, 1133.8) | 61.1 (20.5, 125.6) | 474.2 (144.3, 1102.4) |
| 18-year-olds, 50% coverage | 2.0 (1.4, 2.7) | 39.0 (11.3, 81.4) | 119,851 (89,246, 170,901) | 320.9 (86.4, 743.3) | 35.3 (7.8, 77.6) | 290.1 (58.3, 723.1) |
| **India** | | | | | | |
| *Pre- and post-infection efficacy* | | | | | | |
| 10-year-olds, 80% coverage | 15.2 (12.8, 18.7) | 232.5 (−714.6, 1646.4) | 5,936,801 (4,284,243, 8,609,081) | 38.1 (−106.4, 312.9) | −174.6 (−1528.9, 1392.0) | Dominant −28.1 (−232.9, 255.9) |
| 15-year-olds, 80% coverage | 21.8 (18.3, 26.4) | −694.1 (−1891.0, 853.5) | 9,905,887 (7,073,810, 14,271,184) | Dominant −70.4 (−172.5, 93.74) | −1349.4 | Dominant −134.6 (−330.4, 45.1) |
| 18-year-olds, 50% coverage | 14.5 (12.2, −17.6) | −486.9 (−1255.7, 438.3) | 6,416,849 (4,634,098, 9,148,222) | Dominant −73.5 (−176.3, 72.8) | −885.6 (−2180.9, 123.3) | Dominant −135.1 (−324.4, 19.6) |
| *Post-infection efficacy only* | | | | | | |
| 10-year-olds, 80% coverage | 0.7 (0.5, 0.9) | 1526.6 (714.7, 2885.6) | 404,065 (252,210, 667,108) | 3736.9 (1493.8, 8252.1) | 1506.4 (694.9, 2860.6) | 3 690.8 (1415.2, 8191.6) |
| 15-year-olds, 80% coverage | 1.7 (1.2, 2.4) | 1353.9 (510.7, 2763.9) | 894,399 (542,021, 1,471,362) | 1513.1 (499.9, 3733.2) | 1290.4 (412.7, 2687.5) | 1441.0 (391.4, 3676.0) |
| 18-year-olds, 50% coverage | 1.7 (1.3, 2.3) | 813.3 (225.0, 1651.1) | 896,801 (565,477, 1,429,151) | 917.0 (231.3, 2269.3) | 736.8 (142.4, 1573.8) | 822.4 (143.8, 2172.5) |

Figures represent median, and range (incidence rate reduction) or 5th and 95th percentile values (all others) from 1000 calibrated model runs. Incidence rate reduction is the percentage reduction in tuberculosis incidence in 2050 in scenarios with vaccination compared to the counterfactual scenario without vaccination. Costs estimated from a health service perspective. All costs and DALYs discounted at 3%/year. Cost effectiveness thresholds—South Africa: $2480/DALY averted, India: $264/DALY averted.
DALY disability-adjusted life year, USD United States Dollars.

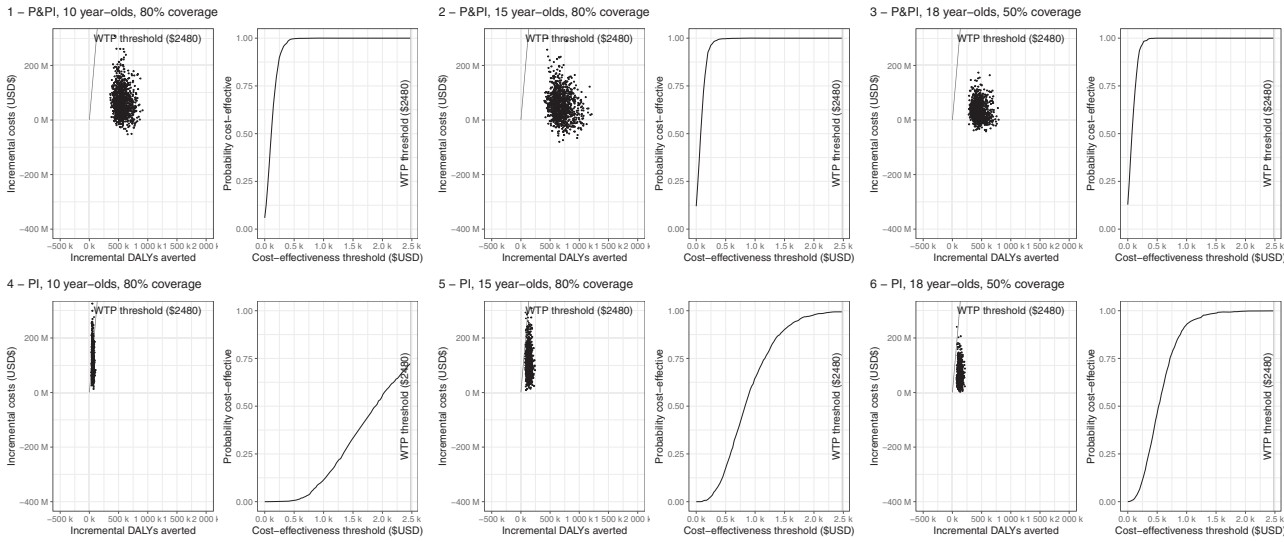

**Fig. 1 Cost effectiveness planes and cost-effectiveness acceptability curves for vaccination scenarios in South Africa.** Top row (panels 1–3) shows cost-effectiveness planes (scatter plot) and cost-effectiveness acceptability curves (line graph) for three scenarios with pre- and post-infection vaccine efficacy. Bottom row (panels 4–6) shows cost-effectiveness planes and cost-effectiveness acceptability curves for three scenarios of post-infection-only vaccine efficacy. P&PI denotes vaccine with pre- and post-infection. PI denotes vaccine with post-infection efficacy only. DALY disability-adjusted life year, USD United States Dollars, WTP willingness to pay.

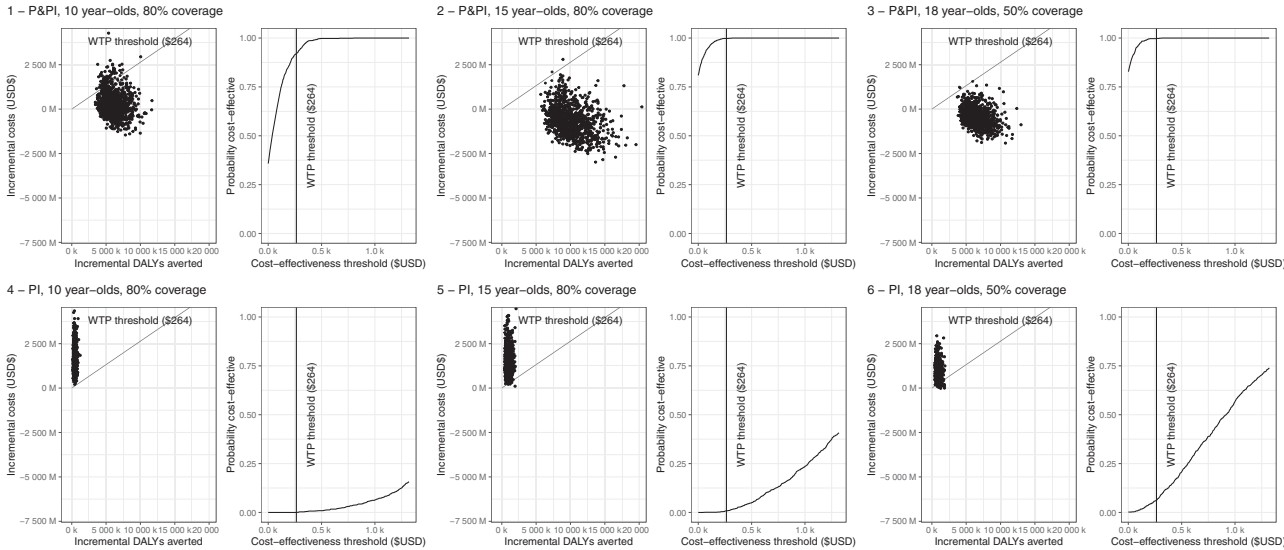

**Fig. 2 Cost effectiveness planes showing incremental cost-effectiveness ratios for vaccine introduction scenarios in India.** Top row (panels 1–3) shows cost-effectiveness planes (scatter plot) and cost-effectiveness acceptability curves (line graph) for three scenarios with pre- and post-infection vaccine efficacy. Bottom row (panels 4–6) shows cost-effectiveness planes and cost-effectiveness acceptability curves for three scenarios of post-infection-only vaccine efficacy. P&PI denotes vaccine with pre- and post-infection. PI denotes vaccine with post-infection efficacy only. DALY disability-adjusted life year, USD United States Dollars, WTP willingness to pay.

health system costs. Notably, around 95% of incremental health service costs were attributable to increases in the cost of ART, as vaccines prevented premature tuberculosis mortality among people living with HIV.

In India, we estimate that routine vaccination with pre- and post-infection efficacy is likely to be cost-effective from the health system perspective (92–100% probability of cost-effectiveness), however, it is unlikely a vaccine with post-infection efficacy only will be cost-effective (0–6% probability of cost-effectiveness) (Table 1, Fig. 2). The pre- and post-infection vaccination scenarios for 15- or 18-year-olds were both cost-saving from a health system perspective and therefore dominant. When patient costs

were also included (Table 1, Supplementary Fig. 9), a vaccine with pre- and post-infection efficacy became cost-saving for all scenarios, while the likelihood that a vaccine with post-infection efficacy was cost-effective doubled to 12% in the most cost-effective scenario.

If the vaccine introduced were post-infection only the most cost-effective implementation scenario of those explored would be among 18-year-olds at 50% coverage—in South Africa this scenario cost $320.9/DALY averted (95% CrI: $86.4, $743.3), and in India $917/DALY averted (CrI: $231.3, $2269.3) from the health system perspective. If the vaccine introduced were a pre- and post-infection vaccine, the 15-year-old scenario with 80%

coverage was the most cost effective in South Africa ($20.6/DALY averted, CrI: $−56.2, $143.1), and the 18-year-old scenario with 50% coverage was most cost saving in India ($−73.5/DALY averted, CrI $−176.3, $72.8). Notably, with both vaccine types and in both settings, a vaccination programme reaching a lower (50%) coverage of 18-year-olds had very similar impact and cost-effectiveness as reaching a higher (80%) coverage of 15-year-olds. These scenarios may have different real-world feasibility depending on the proportion of 15- and 18-year-olds who can be reached by vaccination programmes, for example through education or employment.

In both countries, cost-effectiveness findings did not change in scenarios assuming higher vaccine delivery costs (Supplementary Figs. 10 and 11). Duration of protection was not an important determinant of cost-effectiveness in South Africa, only one vaccination scenario became cost-ineffective with a 5- or 10-year duration of protection. In India, a vaccine with five-year duration of protection was not cost-effective in any scenario, while vaccinating 10-year-olds became cost-ineffective when a 10-year duration of protection was assumed (Supplementary Figs. 12 and 13).

## Discussion

This is analysis used M72/AS01$_E$ efficacy data to explore the cost-effectiveness of routine adolescent vaccination. We estimate that routine adolescent vaccination with a M72/AS01$_E$-like vaccine would be cost-effective in South Africa, and would be cost-effective and potentially cost-saving in India if it provided efficacy pre- and post-infection. We also find that, in both settings, vaccinating fewer 18-year-olds (50%) through routine vaccination is as cost-effective as vaccinating 80% of 15-year-olds. We observe a similar effectiveness in 50% coverage scenarios among 18-year-olds to 80% coverage among 15-year-olds. This is because, in both settings, incidence rates rise steeply between 10 and 19 years of age, however, in South Africa rates increase to a peak at a later age (35–44 years) than India (20–24 years). Therefore, even though coverage is lower among 18-year-olds, in the 15-year duration of protection there are more avertible cases in 18–32-year-olds than 15–29-year-olds. These findings are important to inform decisions of where and how phase III trials of vaccines could be best conducted to maximise value, and how vaccines could be implemented most efficiently.

Implications for clinical trial design include the need for inclusion of adolescent populations in phase III studies to provide countries the option of routine adolescent vaccination upon registration, and the inclusion of both pre- and post-infection populations in studies of these younger age groups given the importance of the additional value from pre-infection protection in adolescents in India. For implementation, our findings demonstrate that adolescent routine vaccination without mass campaigns is likely to be cost effective in these settings, and that higher coverage in 15-year-olds may be similarly cost effective to lower coverage in 18-year-olds, providing implementation flexibility without impacting cost effectiveness in these countries. Our results also demonstrate the importance of including patient costs in such analyses, as several scenarios became cost saving once these costs were accounted for. In addition, inclusion of HIV-related costs in South Africa was important for understanding the overall health system perspective, as they made up the majority of incremental costs.

Notably, a vaccine with post-infection efficacy is likely to be cost-effective in South Africa but unlikely to be cost-effective in India due to two key factors. First, the cost-effectiveness threshold is almost ten times lower in India than South Africa, meaning that, all else equal, scenarios are less likely to be cost-effective

unless net costs are also proportionally lower. Second, there are key epidemiological differences between the two settings. As demonstrated previously[10], the South African epidemic is more relapse/reactivation driven than the Indian epidemic, meaning post-infection efficacy is more important in South Africa while pre-infection efficacy is more important in India.

A strength of this study is that it uses efficacy data from the M72/AS01$_E$ trial to parameterise vaccine impact in dynamic transmission models fitted to tuberculosis epidemics in two countries, South Africa and India, facing substantial tuberculosis burden. In addition, our estimates include HIV-associated costs and patient costs, which were shown to be important contributors to cost-effectiveness estimates. We also explore feasible implementation scenarios, finding that achieving a lower coverage among older recipients (18-year-olds) gives comparable impact to higher coverage among younger recipients (15-year-olds). This is informative as the latter scenario may be more achievable in reality through school-based vaccination programmes, for example in concert with routine human papillomavirus (HPV) vaccination.

One previous paper used M72/AS01$_E$ results to estimate the epidemiological impact of a vaccine with post-infection efficacy on the MDR-tuberculosis epidemic[9], and one other considered the cost-effectiveness of such a vaccine in China and India[11], but both studies assessed the impact of routine adolescent vaccination accompanied by regular all-adult mass vaccination campaigns. No previous studies have explored the cost-effectiveness of only routine adolescent vaccination, which would be much more straightforward to implement through existing platforms than recurrent large all-adult mass campaigns, so may be an attractive option for TB vaccine implementation in resource-limited settings. This work also extends beyond the former study by assessing cost-effectiveness and modelling a vaccine offering pre- and post-infection efficacy. We also extend beyond the latter study by estimating the cost-effectiveness of such a vaccine in South Africa in the presence of the HIV syndemic, and—in both settings—taking a more sophisticated approach to targeting among adolescents and by including patient costs in the estimates. Our estimates that a M72/AS01$_E$-like vaccine would be cost-effective in South Africa are consistent with previous cost-effectiveness studies parameterised for hypothetical vaccines[6,12]. However, our finding that an adolescent-only routine vaccination scenario with post-infection-only efficacy is unlikely to be cost-effective in India, even when a societal perspective is taken, is timely and highlights the importance of trials exploring both pre- and post-infection efficacy in adolescents in this high-burden setting.

This study has a number of limitations. As in any model-based analysis of technologies in development, we make assumptions about vaccine characteristics and implementation. The model takes a 26-year time horizon which means that benefits accruing after 2050 are not estimated. In addition, the proportion of tuberculosis infections which are multidrug-resistant is assumed to be constant over this period. Future trends of MDR-tuberculosis are unknown, but if this proportion were to increase then the comparatively high costs of treating MDR-tuberculosis mean we may underestimate the cost-effectiveness of vaccines. In South Africa, we assume ART scale-up aligns with the 90:90:90 goals. UNAIDS data from 2020 indicates that the first and third "90" have been achieved[13], however, only 75% of people aware of their HIV-positive status are on ART. If fewer people are on ART in 2025 than we assume, HIV-related mortality will be higher and therefore fewer tuberculosis cases will be averted by a vaccine among people living with HIV, though ART costs would also be lower. Few data are available for TB vaccination in HIV-positive populations, so reduction in efficacy in

**Table 2 Characteristics of modelled M72/AS01$_E$-like vaccines.**

|                              | Scenario 1                    | Scenario 2                    | Scenario 3                    | Scenario 4                 | Scenario 5                 | Scenario 6                 |
| ---------------------------- | ----------------------------- | ----------------------------- | ----------------------------- | -------------------------- | -------------------------- | -------------------------- |
| Efficacy by infection status | 50% pre and post infection    | 50% pre and post infection    | 50% pre and post infection    | 50% post infection only    | 50% post infection only    | 50% post infection only    |
| Vaccination age              | 10 years                      | 15 years                      | 18 years                      | 10 years                   | 15 years                   | 18 years                   |
| Coverage                     | 80%                           | 80%                           | 50%                           | 80%                        | 80%                        | 50%                        |

In all scenarios, duration of protection was 15 years and vaccine implemented models run for 26 years in the period 2025–2050 assuming a per-person cost of vaccination of $5.

this population was assumed based upon data from other vaccines, which was likely a conservative assumption, leading to a possible underestimate of cost effectiveness in South Africa. Efficacy in HIV-positive populations should be explored in future work once data in this population become available. In addition, there is considerable uncertainty about the duration of protection offered by the M72/AS01$_E$ vaccine. Trials demonstrated three years of sustained antibody response, so mathematical models such as this have used expert opinion suggesting that 10 or 15 a year duration of protection is a reasonable assumption. However, real-world evidence is needed to confirm this; if the duration were shorter, results for older adolescent protection would likely be more cost-effective than estimated here.

We did not model the detailed implementation of routine vaccination and may therefore omit dynamics which could improve or lessen the cost-effectiveness of M72/AS01$_E$ introduction. Importantly, the health service capacity to absorb a new campaign and associated new activities are likely to be constrained in financial, human resource or policy domains[14]. We do not estimate costs of expanding capacity in physical or human resources to deliver vaccines, potentially underestimating implementation costs, though a future TB vaccination programme could feasibly be combined with existing adolescent HPV vaccination, for example. However, future reductions in tuberculosis burden would lead to lower health service use, potentially freeing up capacity for other activities. It is likely that savings from the latter would be greater than costs from the former, which would make vaccines more cost-effective. Similarly, we assume that there are no changes to other TB prevention and control measures over the period modelled. Finally, we assume that direct costs to patients (i.e. omitting earning losses) collected among adult TB patients are the same among adolescents. In reality, the cost of paediatric TB disease may be higher than we assume here if, for example, direct and indirect costs of carers were considered, making our societal estimates of cost-effectiveness conservative.

Our main finding, that routine adolescent vaccination with a M72/AS01$_E$-like vaccine which is efficacious either post-infection-only or pre- and post-infection would be a cost-effective intervention in South Africa, and a vaccine with pre- and post-infection efficacy is likely to be cost-effective in India, strengthens the case for continued investment in M72/AS01$_E$ vaccine development. For adolescent vaccination in India, it will be important to understand if and how much M72/AS01$_E$ provides efficacy in pre-infection populations. It is critical that future development work engages both high-tuberculosis burden countries where vaccines would be introduced, alongside potential adolescent or young adult vaccine recipients, to ensure that vaccine characteristics and delivery mechanisms meet their preferences and needs.

## Methods

**Epidemiological assumptions**. A full description of the epidemiological model has been published previously[10], with model structure and parameterisation detailed on page 7 of the previous publication, and demographics, calibration and analysis on page 8. A brief description is also provided in the supplementary information for this study. In summary, we use an age-structured compartmental dynamic *M.tb*

transmission model to assess the population-level impact and cost-effectiveness of a new M72/AS01$_E$-like vaccine, delivered during 2025–2050 to adolescents in South Africa and India. India was selected due to contributing the largest number of TB cases globally (26% of prevalent cases in 2019), and South Africa as it reports one of the highest national rates of TB globally, in addition to being the eighth highest in absolute terms and reports an important level of HIV co-infection. Demographics were parameterised using UN population division data and projections, HIV in South Africa was parameterized using age- and year-specific HIV incidence and AIDS-related mortality from Spectrum data and projections, and ART was parameterised historically using data from Spectrum, scaled up to 90% coverage by 2022 to meet the 90:90:90 targets, and held at 90% coverage beyond 2022. The model was calibrated to tuberculosis prevalence, incidence, mortality, and notification data, age- and HIV-stratified where available, for South Africa and India. Natural history uncertainty was captured in 1000 calibrated model runs.

**Vaccine assumptions**. We modelled a range of vaccine characteristics and introduction scenarios in each country (Table 2), varying routine vaccination age (10-, 15- or 18-year-olds), and whether a vaccine was efficacious post-infection or pre- and/or post-infection. The choice of vaccination ages was driven by epidemiological and logistical factors. Vaccination of 10-year-olds assumed platform sharing and potential co-administration with HPV vaccination. Vaccination of 15-year-olds aimed to vaccinate older teens in which TB incidence is increasing[15], but yet school attendance remains high, to ensure good coverage through school-based vaccine delivery[16]. Vaccination of 18-year-olds provides the opportunity to provide protection to young adults as TB incidence rates are increasing, extending through towards the epidemiological peak[17]. Scenarios assumed 80% coverage in 10- or 15-year-olds, or 50% coverage in 18-year-olds (assuming they were harder to reach). To represent a real-world routine vaccination programme, we assume no pre-screening for tuberculosis status or requirement for prior receipt of BCG before vaccination. We assume immediate production, scale up, and delivery from 2025, and that no mass or catch-up vaccination campaigns take place.

Based on the phase 2B primary efficacy results, we assume 50% vaccine efficacy against disease[3]. Vaccine was assumed safe in HIV-positive populations, with a 20% relative reduction in efficacy compared to HIV-negative populations. Duration of protection was assumed as 15 years waning instantly at the end of the protection in the base case, though we also explore 5- and 10-year duration of protection in sensitivity analyses. Two types of vaccine efficacy by host infection status are modelled to reflect uncertainty in the population likely to benefit from the vaccine: post-infection which only protects when delivered to latently infected or recovered hosts, and pre- and post-infection, which protects when delivered to uninfected or infected hosts.

Epidemiological and health economic outcomes were estimated per model run as the difference between the no-new-vaccine baseline scenario and each vaccination scenario. Outcomes are presented as medians and credible intervals from the 1000 model runs.

**Costs**. We used an ingredients-based approach to estimate the net cumulative costs of routine vaccination from a health service perspective, including vaccination costs incurred and tuberculosis services costs averted. In an additional analysis, we take a societal perspective and include patient-incurred costs of tuberculosis treatment. Adolescent patient cost data were not available, so we assumed that patient-incurred direct medical and non-medical costs were the same as those faced by adults and omit indirect costs which could feasibly differ among adolescents, in particular income losses. Given the close relationship between HIV and tuberculosis epidemics in South Africa, we also estimated additional costs of HIV treatment as a result of increased life-expectancy due to reduced tuberculosis mortality. In the South Africa epidemiological model, we assume that antiretroviral therapy scale-up is consistent with UNAIDS 90–90–90 targets. Vaccine purchasing and delivery costs were unknown, so were drawn from a distribution of Gamma(6, 0.83) equating to an average of $5 per person, based on information provided by potential funders.

Due to the considerable uncertainty in estimating the vaccine price for a vaccine that does not yet exist, and the costs of delivering such a vaccine to adolescents, we conducted a separate analysis assuming a feasible higher vaccine cost. We reviewed a range of sources of vaccine delivery costs and prices, and constructed a higher vaccine cost scenario, assuming that the vaccine had the same price as the GAVI/UNICEF negotiated HPV vaccine without GAVI support ($4.60), and the highest

vaccine delivery cost estimate we located ($2.93), drawing uncertainty bounds around the resultant central vaccine cost estimate of $7.53 to draw from a higher cost distribution of Gamma(6.05, 1.24).

We estimate the health economic impact of reductions in mortality and morbidity by estimating the cumulative number of disability-adjusted life years (DALYs) incurred by tuberculosis disease using DALY weights and accounting for HIV co-infection. MDR-tuberculosis is an important driver of tuberculosis programme costs. Due to the relative stability of the global MDR-tuberculosis epidemic, we model MDR-tuberculosis as a constant proportion of all tuberculosis: 6.2% in India[18] and 4.6% in South Africa[19]. We assume rifampicin-resistant tuberculosis received MDR regimens as it is generally treated with the same drugs for the same duration as MDR-tuberculosis, including in South Africa[20]. We assume no changes to other prevention and control measures over the future time horizon modelled.

We conducted a probabilistic sensitivity analysis to explore the impact of epidemiological and economic parameter uncertainty on results, where results from 1000 fitted epidemiological model runs were integrated with 1000 draws from cost distributions, assuming uncertainty in all cost parameters was represented by gamma distributions. Costs and DALYs were discounted by 3% per year as standard[21]. We assess cost-effectiveness by comparing the incremental cost/DALY averted by vaccines to conservative, lower-bound supply-side country-specific cost-effectiveness thresholds—$264/DALY averted in India and $2 480/DALY averted in South Africa[22,23]. Economic outcomes are summarised as incremental cost per DALY averted.

**Cost-effectiveness thresholds**. The choice of cost-effectiveness threshold is critical to a cost-effectiveness analysis, and represents the maximum willingness-to-pay of a payer (usually a health system) for one unit of health[24]. Thresholds typically represent the marginal disinvestment required by a health system to invest in a new technology under a limited budget—representing the opportunity cost of spending, these are termed supply-side thresholds. Thresholds should be ideally calculated at a country level, and reflect context-specific heterogeneity and disinvestment decisions, however, this is not often possible due to data or analytic constraints. Until recently, it was common for analyses to use 1x or 3x GDP/capita as a threshold in the absence of an empirical estimate, citing the World Health Organization's Commission of Macroeconomics and Health[25]—yet it was never intended for these figures to be used as cost-effectiveness thresholds, and in practice produced thresholds which were unrealistically high[26].

Instead, we follow recent literature in cost-effectiveness thresholds, which takes cost-effectiveness thresholds in settings where they can be reliably estimated and extrapolates to other settings based on convertible metrics. We use the estimates of Ochalek et al.[22]—these are generally lower and therefore more conservative than taking a GDP/capita threshold. Ochalek et al. present four thresholds for each of a number of countries, and variations in calculation method mean they vary in magnitude; we take the lowest and therefore most conservative of the four estimates for South Africa ($2480/DALY averted) and India ($264/DALY averted).

**Reporting summary**. Further information on research design is available in the Nature Research Reporting Summary linked to this article.

## Data availability
Epidemiological and health economic analyses were conducted as described in the manuscript based on publicly available data collated in the supplementary appendix. All data used for parameterisation or calibration in this study are publicly available and referenced. Data generated for the main outcomes of this study are in the tables and figures of the main manuscript or the supplementary materials of this paper.

## Code availability
Epidemiological models used in this study were as previously described in the publication Harris et al. 2020 Science Translational Medicine. The epidemiological and economic modelling code is available here https://doi.org/10.5281/zenodo.5793303.

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

## Acknowledgements
This work was funded by the Bill and Melinda Gates Foundation (Opportunity ID: OPP1160830).

## Author contributions
R.H., C.W., G.G. and R.W. conceived the study. R.H., C.W., T.S. and R.W. developed the epidemiological models, and M.Q., G.G. and F.B developed the economic models. R.H., M.Q., C.W. and F.B. curated research data and performed the analysis. T.S. and G.G. provided validation of study findings and input on model design. M.Q. and R.H. prepared the first draft of the manuscript, and all authors contributed to the revised manuscript.

## Competing interests

R.H. and G.G. report current employment at Sanofi Pasteur, but do not work on TB or TB vaccines in their roles. R.H., G.G., C.W. and R.W. report grants from BMGF for the conduct of the study. C.W. received funding from UKRI/MRC MR/N013638/1. Funders had no role in study design, data collection, data analysis, data interpretation, or writing of the report. All other authors declare no competing interests.
