## [Peer Review File · Nature Communications]

Cost-effectiveness of routine adolescent vaccination with an M72/AS01_E-like tuberculosis vaccine in South Africa and IndiaREVIEWER COMMENTS

Reviewer #1 (Remarks to the Author):

I commend the authors for this is an interesting and valuable study. I have a few comments that may be helpful:

1. Because this a cost-effectiveness paper, the results hinge on the cost assumptions. I find the sentence “Vaccine purchasing and delivery costs were unknown, so were drawn from a distribution of Beta(6,0.83) equating to a mean of \$5 per person, based on information provided by potential funders.” on page 12 a bit inadequate. Please clarify who are these “potential funders” and what type of information (including source) they provided.

2. South Africa has a national adolescent vaccination program, delivering Tetanus and diphtheria vaccines at ages 6 and 12. India has no adolescent vaccination program and the routine program can only vaccinate 83% of under-2 children. Is it realistic to simply assume \$5 per vaccinated adolescent cost in these contexts? How will the programs actually introduce the vaccine (routine/ campaign) and at what cost? Some discussion on this is necessary in my opinion.

3. The database <http://immunizationeconomics.org/ican-idcc> has unit costs of delivering vaccines in India, South Africa, and many other countries. A field study estimated cost of delivering routine childhood vaccines in 7 India states. <https://gh.bmj.com/content/3/3/e000794.abstract> . Should these data not be used?

4. The COVID vaccination program in India may give some idea about the potential cost of an adolescent/ adult vaccine. The Indian government estimated INR 700 per person cost for a 2 dose vaccine. <https://theprint.in/economy/rs-35000-cr-covid-vaccine-allocation-in-budget-can-cover-50-cr-indians-expenditure-secy/597542/> . The vaccine is being sold to the government for INR 150 per dose, implying INR 400 in delivery cost <https://www.livemint.com/news/india/covishield-to-cost-400-per-dose-for-states-600-for-private-hospitals-11618991072705.html>.

Reviewer #2 (Remarks to the Author):

Note that there is a mistake in the second sentence of the Results section where it begins “A vaccine with pre- and post-vaccination ...”. It should read “pre- and post-infection...”.

The major claims (paraphrasing) of the paper include the following:

All adolescent routine vaccination scenarios substantially reduce TB incidence rates in 2050.

A vaccine with a pre- and post-infection efficacy had a much greater impact (on incidence and DALYs averted) than did a vaccine with only post-infection efficacy. This difference was greater in India than in South Africa.

Routine adolescent vaccination with a M72/AS01E-like vaccine would be cost-effective in South Africa, and would be cost-effective and potentially cost-saving in India if it provided efficacy pre-infection as well as post-infection.

Including the patient costs (to reflect a societal perspective of cost effectiveness) made all scenarios more cost-effective and even cost-saving for a vaccine with both pre- and post-infection efficacy.

Vaccinating fewer 18-year-olds (50%) through routine vaccination is as cost-effective as vaccinating 80% of 15 year-olds.

The claims are indeed novel and will be of interest to others particularly to those in the TB and vaccine community. They fill a gap in the existing literature, in particular by informing the cost-effectiveness of TB vaccination of adolescents that could be part of routine immunization platform. It also investigates the important distinction of cost-effectiveness among those without TB infection in addition to those with TB infection. Furthermore, it includes patient costs for a more societal perspective of cost benefit.

The conclusions are indeed original and the work is indeed convincing. There are a few things that could strengthen the paper if not the conclusions. One can presume that South Africa and India are chosen to contrast the impact of a vaccine in the epidemiology of TB in a country with and without HIV. However, the specific impact of HIV on the different results in the two countries is not clear.

What is it about the vaccine characteristics, TB epidemiology or the economics that result in country differences in cost effectiveness? The value of the study would be enhanced if it gave an intuitive explanation of the reasons underlying the divergent results in the two countries:

- Why is pre-infection protection from vaccine more important in India than South Africa. Or in other words, why is post-infection efficacy only cost-effective in South Africa and not India? Is it to do with the differences in age-specific infection, HIV, or the cost-effectiveness thresholds?

- Why is it that lower coverage in 18 year olds is as effective as a higher coverage in 15 year olds?

Providing these insights from the model can enhance the generalizability of this work beyond South Africa and India. While this study is specific to two countries, the insights behind these country differences can be informative to more countries or regions within countries with epidemiology analogous to either of the two countries.

Furthermore, more justification for the particular age cutoffs employed would be helpful in particular if it helps to align policy implications with HPV vaccination; also as it relates to value of BCG vaccination which is thought to wane at around age 12.

The methods are appropriate and valid.

This paper is important as it will aid in prioritizing the development of TB vaccines most likely to reach the 2050 TB elimination goal. It is also a salient article in the use of mathematical models for public health policy.

Reviewer #3 (Remarks to the Author):

The authors present a cost-effectiveness analysis for country-level rollout (South Africa and India in adolescents) of a prevention of disease and/or prevention of infection TB vaccine roughly parameterized by the phase 2b trial of M72/AS01E. The calibrated models on which the study is based are published in peer-reviewed literature and referenced appropriately. Overall, the paper is well written, clear in its conclusions and it highlights the need to consider both prevention of infection and prevention of disease vaccines in different epidemiological settings. An area which would benefit from further clarification are the assumptions around future demographics and non-TB death rates, given the long time horizon of 2050 (26 years of intervention). As well, as the cost-effectiveness results will be strongly dependent on duration of protection, the paper would benefit from some sensitivity analysis around this. It would also benefit from moving discussion of cost-effectiveness thresholds from the supplementary material to the main text.

Specific points follow below:

1. In the intervention scenarios a 15 year duration of protection is assumed based on expert opinion. As noted, efficacy trials are 3 years in duration. It would be helpful to expand on any evidence or at least a hypothesis for this assumption (eg. decay rate of T-cell responses etc). As well given the uncertainty here, some additional sensitivity analysis would be informative, such as what is the minimum duration of protection to be cost-effective under the different thresholds? (slightly beyond scope of the paper perhaps) But at what point does this push things to a place where targeting adult pops might be cost-effective but not adolescents (considering realistic coverage levels for groups).

2. As the time horizon is 26 years, even considering discounting, changes in future death rates can impact estimates of DALYs averted. A short description of how non-TB death rates are extrapolated to the future would make this transparent. Are non-TB death rates extrapolated linearly in terms of current trends, or with some diminishing returns, or just held constant. What are assumptions around both HIV specific death as well as ART coverage, HIV prevalence. A concise statement of these with a reference to the calibrated model would suffice. It is noted there is discussion of HIV assumptions and 90-90-90 targets in the Discussion section but including this in the methods would be useful.

3. Moving some of the section on how CE thresholds are set from supplementary material to the main text would be helpful to the readers to understand the reasoning behind them.

4. A 20% relative reduction in efficacy is assumed in HIV positive relative to HIV negative populations. Currently there is no efficacy trial data for this. Does this assumption implicitly take into account levels of viral suppression?. If so, shouldn't this change in time based on assumptions around ART coverage and viral suppression?

5. It would be more clear in the table if efficacy were split into pre and post and the numbers listed. I take it the sum of efficacies is the same (ie pre + post) across the scenarios for comparison?

Minor points:

“When patient costs were also included (societal perspective), all scenarios became more cost-effective .. “ The wording in this case could be ambiguous since this is true in the definitional sense, it could read like a result.

Reviewer 1

I commend the authors for this is an interesting and valuable study.

Response:

We thank the reviewer for these positive comments.

I have a few comments that may be helpful:

1. Because this a cost-effectiveness paper, the results hinge on the cost assumptions. I find the sentence “Vaccine purchasing and delivery costs were unknown, so were drawn from a distribution of Beta(6,0.83) equating to a mean of \$5 per person, based on information provided by potential funders.” on page 12 a bit inadequate. Please clarify who are these “potential funders” and what type of information (including source) they provided. South Africa has a national adolescent vaccination program, delivering Tetanus and diphtheria vaccines at ages 6 and 12. India has no adolescent vaccination program and the routine program can only vaccinate 83% of under-2 children. Is it realistic to simply assume \$5 per vaccinated adolescent cost in these contexts? How will the programs actually introduce the vaccine (routine/ campaign) and at what cost? Some discussion on this is necessary in my opinion. The database <http://immunizationeconomics.org/ican-idcc> has unit costs of delivering vaccines in India, South Africa, and many other countries. A field study estimated cost of delivering routine childhood vaccines in 7 India states. <https://gh.bmj.com/content/3/3/e000794.abstract> . Should these data not be used?

The COVID vaccination program in India may give some idea about the potential cost of an adolescent/ adult vaccine. The Indian government estimated INR 700 per person cost for a 2 dose

vaccine. <https://theprint.in/economy/rs-35000-cr-covid-vaccine-allocation-in-budget-cancel-over-50-cr-indians-expenditure-secy/597542/> . The vaccine is being sold to the government for INR 150 per dose, implying INR 400 in delivery cost <https://www.livemint.com/news/india/covishield-to-cost-400-per-dose-for-states-600-for-private-hospitals-11618991072705.html>.

Response:

We would like to sincerely thank the reviewer for their careful reading of the paper and considered comments on the vaccine cost assumptions. These assumptions are indeed key drivers of model results and the comments and references they provide are helpful making model results more robust.

The \$5 per dose cost assumptions came from the Bill and Melinda Gates Foundation, who in 2020 acquired the licence to trial and develop the M72 vaccine, and who funded this work. There is considerable uncertainty in estimating a) the price of vaccine doses for a vaccine which doesn't yet exist, and b) the costs of delivering such a vaccine to adolescents when, as the reviewer points out, such a programme does not exist in India, and South Africa's DTP programme only reaches 12-year-olds, and not older ages of 15 and 18 year olds as modelled in this study. Given this uncertainty, we sought input from the Foundation on the

target price and health system cost of the M72 vaccine (\$5 per dose), which was used in the submitted analysis.

Guided by the reviewer's comments, we have identified a number of plausible vaccine price and cost assumptions, summarised in table R1, which include the helpful sources identified by the reviewer.

We also noticed a typo in the paper, the cost distributions listed as beta distributions were in fact gamma distributions (as standard practice given the skew in cost data - beta distributions are generally used in PSAs for variables bounded 0,1). We have updated this in the paper.

We note that almost all combinations of vaccine delivery cost and vaccine price for South Africa and India summarised in table R1 below result in a lower cost-per-dose than the \$5 assumed in the base case. In addition, the gamma distribution of the vaccine cost used in the base case probabilistic sensitivity analysis (Figure R1) covers the full range of possible vaccination costs identified.

Figure R1: Base case vaccine cost density curve

Table R1: Possible vaccine price and cost assumptions

	South	India	Notes	Source
Bill and Melinda Gates Foundation (expert opinion)	\$ 5	\$5		
Assumptions of vaccine price borne by country and delivery costs, and variation in this based on variation of HPV delivery costs from literature	Lower bound \$1, Upper bound \$9	Lower bound \$1, Upper bound \$9		
Vaccine delivery costs only				
Empirical data on vaccine delivery costs (excluding vaccine price, India only)	US\$ 1.66 per dose supplied	US\$ 1.66 per dose supplied	"Cost per dose delivered inclusive of vaccine cost varied from US\$1.38 across regions for in Bihar to US\$2.93 in Kerala". Price per dose was under \$0.20 for all infant but Hib vaccinations, which was \$2.11 and only conducted in Gujjurat vaccination) and Kerala.	https://gh.bmj.com/content/3/e00794.abstract
Total financial costs of delivering COVID-19 vaccine (global)	Global estimate: \$0.99 Lower bound: \$1.45 Upper bound: \$2.54	US\$ 1.66 per dose supplied	WHO/COVAX: https://www.who.int/docs/default-source/coronavirus/act-accelerator/covax/costs-of-covid-19-vaccine-delivery-in-92amc-08.02.21.pdf	
Immunization Delivery Cost Catalogue (immunizationeconomics.org)	—	—	The database does not identify sufficient cost data from South Africa estimate of to report a cost per dose or fully-immunised person. For India, three adolescent studies estimate cost per dose (lowest - highest \$0.41-\$0.50) and two (vaccination) estimate cost of adolescent vaccination (\$1.91 and \$2.81). From reviewer: "The Indian government estimated INR 700 per person cost for a 2 dose vaccine.. The vaccine is being sold to the government for INR 150 per dose, implying INR 400in delivery cost."	http://immunizationeconomics.org/ican-idcc
COVID-19 Vaccine delivery cost (Inferred from media reports)	\$2.73 per dose	\$2.73 per dose	We halve this INR 400 cost since the M72 is a one-dose vaccine, giving IN R200= \$2.73 USD	https://theprint.in https://www.livemint.com
Vaccine price				
Price to country of GAVI-acquired HPV vaccines	Initially \$.20, rising to \$0.40 \$4.60 after five years of "transition phase"	Initially \$.20, rising to \$0.40 \$4.60 after five years of "transition phase"	Initially \$.20, rising to \$0.40 after five to \$0.40 after five years of But, in early stages of rollout, countries only pay \$0.20 per dose, "transition increasing by 15% per year in "transition phase". Price here assumed phase" after five years of transition phase.	https://www.gavi.org/types-support/vaccine-support/human-papillomavirus

Nevertheless, to be conservative, we identified the highest possible vaccine delivery cost based on these data. This was if the M72 vaccine had the same price as the GAVI/UNICEF negotiated HPV vaccine without GAVI support (\$4.60), and if delivery cost \$2.93 (based on Indian data as the \$1.66 for South Africa is lower and not based on South African data).

We modelled the highest central estimate cost of $\$4.60 + \$2.93 = \$7.53$ in South Africa and India. We ran a full probabilistic sensitivity analysis on all scenarios using this increased central cost estimate, taking the wide ratio of the uncertainty range used in the base case, where the gamma distribution parameters were calculated to reflect a central estimate of vaccine cost = \$5, upper bound \$9 lower bound \$1. The uncertainty margin was calculated as $(4/5) * \$7.53 = \6 , uncertainty bounds of the higher price therefore \$1.53-13.53, and the resultant gamma distribution of the higher price scenario $\text{gamma}(6.05, 1.24)$, shown in Figure R2.

Figure R2: Base case vaccine cost density curve - higher cost scenario

The results of this analysis are presented in Figures R3 and R4 below. In short, although individual scenarios' ICERs change as expected and become less cost-effective, there are no changes to the inference of cost-effectiveness (or lack of) made in the base case. For example, in South Africa, the least cost-effective scenario (post-infection-only vaccine with 80% coverage among 10 year-olds) remains over 99% likely to be cost-effective, whilst in India there are no substantive changes to the probability of cost-effectiveness.

We have made the following changes to the paper to show these additional analyses. We have added the following text to the methods section:

“Because of the considerable uncertainty in estimating the vaccine price for a vaccine which doesn’t yet exist, and the costs of delivering such a vaccine to adolescents, we conducted a separate analysis assuming a feasible higher vaccine cost. We reviewed a range of sources of vaccine delivery costs and prices, and constructed a higher vaccine cost scenario, assuming that the vaccine had the same price as the GAVI/UNICEF negotiated HPV vaccine without GAVI support (\$4.60), and the highest vaccine delivery cost estimate we located (\$2.93), drawing uncertainty bounds around the resultant central vaccine cost estimate of \$7.53 to draw from a higher cost distribution of Gamma(60.5,1.24). “

And the following text to the results section, alongside figures 10 and 11 in the Appendix :

“In both countries, cost-effectiveness findings did not change in scenarios assuming the higher vaccine delivery costs (Appendix figures 10 and 11).”

Reviewer 2

The claims are indeed novel and will be of interest to others particularly to those in the TB and vaccine community. They fill a gap in the existing literature, in particular by informing the cost-effectiveness of TB vaccination of adolescents that could be part of routine immunization platform. It also investigates the important distinction of cost-effectiveness among those without TB infection in addition to those with TB infection. Furthermore, it includes patient costs for a more societal perspective of cost benefit.

Response: *We thank the reviewer for their positive feedback and recognition of the value and importance of this research to the TB and vaccine community.*

The conclusions are indeed original and the work is indeed convincing. There are a few things that could strengthen the paper if not the conclusions.

1. One can presume that South Africa and India are chosen to contrast the impact of a vaccine in the epidemiology of TB in a country with and without HIV. However, the specific impact of HIV on the different results in the two countries is not clear.

Response:

We thank the reviewer for this question. Although HIV is an important difference between these two countries, and one of the reasons for choosing them, the key reason for choosing India was because it contributes the largest number of TB cases globally (26% of prevalent cases in 2019) and South Africa because it has one of the highest rates of TB globally (in addition to being the 8th highest in terms of absolute numbers). We have added the justification of choice of country to the methods on page 11:

“India was selected due to contributing the largest number of TB cases globally (26% of prevalent cases in 2019), and South Africa as it reports one of the highest national rates of TB globally, in addition to being the eighth highest in absolute terms and reports an important level of HIV co-infection.”

HIV is a key driver of the epidemic in South Africa (>10% population prevalence and >50% TB-HIV coinfection proportion), so HIV stratification was included in the model. Whereas in India there is a very limited contribution of HIV as a risk factor (<0.5% population prevalence and 3% TB-HIV coinfection proportion) so HIV was not included as a stratification. Other key differences between these two country models include differences in demographics, baseline burden of disease and trends, social mixing patterns, TB control measures, and the India model also accounted for differences in private sector delivery of care. Therefore although some of the difference between the countries could be attributed to HIV and its associated costs, there are many factors contributing to the differences between these two countries, so the specific impact of HIV cannot be isolated.

2. What is it about the vaccine characteristics, TB epidemiology or the economics that result in country differences in cost effectiveness? The value of the study would be enhanced if it gave an intuitive explanation of the reasons underlying the divergent results in the two countries:

- **Why is pre-infection protection from vaccine more important in India than South Africa. Or in other words, why is post-infection efficacy only cost-effective in South Africa and not India? Is it to do with the differences in age-specific infection, HIV, or the cost-effectiveness thresholds?**
- **Why is it that lower coverage in 18 year olds is as effective as a higher coverage in 15 year olds?**

Providing these insights from the model can enhance the generalizability of this work beyond South Africa and India. While this study is specific to two countries, the insights behind these country differences can be informative to more countries or regions within countries with epidemiology analogous to either of the two countries.

Response:

We thank the reviewer for highlighting that we have not explained these points adequately in the manuscript. There is not one sole factor which explains why a pre- and post-infection protection is more likely to result in a cost-effective vaccine in South Africa than India, however the reviewer is correct in their suggestion of potential reasons for this.

First, as the reviewer notes, the probability that scenarios are cost-effective depends on both ICER estimates and the cost-effectiveness thresholds used. South Africa has a much higher GDP/capita than India, and this is used as the key adjustment metric in the study of Ochalek et al. to estimate LMIC cost-effectiveness thresholds. Therefore, all else equal, unless costs are lower in India proportionately to GDP/capita, the higher cost-effectiveness threshold in India means that the same intervention is more likely to be cost-effective in South Africa than in India. This is likely the main contributing factor to the differences we observe in probability of cost-effectiveness between the two settings.

Second, also as noted by the reviewer, there are epidemiological differences between the two settings. As demonstrated in Harris et al 2020, the South African epidemic is projected to be slightly more reactivation/relapse driven over the timeframe of the model, whereas the epidemic in India is strongly more new-infection driven. The post-infection efficacy in South Africa therefore has a greater impact than India due to the relapse/reactivation that becomes avertible in post-infection populations. The addition of pre-infection efficacy has greater impact in terms of percentage rate reduction in India than South Africa as there is a greater proportion of new infections to avert. Analogous to this is that the percentage of the population that is already infected in the paediatric/adolescent age group is slightly lower in the modelled epidemic India (11%) than in South Africa (13%) (Harris et al. 2020), meaning the additional effective coverage of adding pre-infection efficacy is greater in India.

We have added the following text to the discussion section of the manuscript to comment on these two factors:

“Notably, a vaccine with post-infection efficacy is likely to be cost-effective in South Africa but unlikely to be cost-effective in India due to two key factors. First, the cost-effectiveness threshold is almost ten-times lower in India than South Africa,

meaning that, all else equal, scenarios are less likely to be cost-effective unless net costs are also proportionately lower. Second, there are key epidemiological differences between the two settings. As demonstrated previously¹⁰, the South African epidemic is more relapse/reactivation driven than the Indian epidemic, meaning post-infection efficacy is more important in South Africa than India, and pre-infection efficacy is most important in India. ”

In South Africa, there is an almost linear and steep increase in the incidence rate between 10 and 19 years of age (Bunyasi et al. 2021), with incidence rates increasing to a peak in 35-44 year olds. In India, incidence rates increase from around 10 years of age, peaking in the 20-24 year age group (RNTCP, 2019). Therefore, lower coverage of 18 year olds is compensated for as the underlying epidemic curve means that there are more avertible cases in 18-32 year olds than 15-29 year olds (given 15 year duration of protection).

We have added the following to the discussion section to explain this:

“This is because, in both settings, incidence rates rise steeply between 10 and 19 years of age, increasing to a peak in 20-24 year olds in India and 35-44 year olds in South Africa . Therefore, even though coverage is lower among 18 year-olds, in the 15-year duration of protection there are more avertible cases in 18-32 year-olds than 15-29 year-olds.”

3. Furthermore, more justification for the particular age cutoffs employed would be helpful in particular if it helps to align policy implications with HPV vaccination; also as it relates to value of BCG vaccination which is thought to wane at around age 12.

Response:

We thank the reviewer for this suggestion. The ages for vaccination were selected to align with:

- *10 year olds - The age of HPV vaccination in South Africa (9-10year olds) to allow for the same platform for roll out and/or co-administration.*
- *15 year olds - TB rates are lowest in the 10-14 year old age group, with rates in South Africa increasing almost linearly by year from the age of 10y to 19y (Bunyasi et al. 2021). There is 2.6% non-attendance at school for 15yos, compared to 4.6% and 9.8% in 16 and 17 yos, respectively (Statistics South Africa, 2018). Therefore, vaccination of this age group would allow roll out of vaccination through schools in older teens where TB incidence is higher, but at an age before school attendance begins to decline, to maximise coverage.*
- *18 year olds - TB incidence rates are lowest in young children, and increase through teenage year and young adulthood to reach the epidemiological peak in the 35-44 year age group (South African Medical Research Council, 2018). Therefore, vaccination of 18 year olds provides the opportunity to provide protection to young adults as rates are increasing, extending through towards the epidemiological peak.*

To ensure this is clearer in the manuscript, we have added the following text to the vaccine assumptions section on P13 of the manuscript:

“Choice of vaccination ages was driven by epidemiological and logistical factors. Vaccination of 10 year olds assumed platform sharing and potential co-administration with HPV vaccination. Vaccination of 15 year olds aimed to vaccinate older teens in which TB incidence is increasing, yet school attendance remains high for good coverage through school-based vaccine delivery. Vaccination of 18 year olds provides the opportunity to provide protection to young adults as TB incidence rates are increasing, extending through towards the epidemiological peak.”

4. Note that there is a mistake in the second sentence of the Results section where it begins “A vaccine with pre- and post-vaccination ...”. It should read “pre- and post-infection...”.

Response: Thank you - we have fixed the typo to read “A vaccine with pre- and post-infection efficacy...”

The methods are appropriate and valid. This paper is important as it will aid in prioritizing the development of TB vaccines most likely to reach the 2050 TB elimination goal. It is also a salient article in the use of mathematical models for public health policy.

Response: We thank the reviewer for their kind feedback.

Reviewer #3

The authors present a cost-effectiveness analysis for country-level rollout (South Africa and India in adolescents) of a prevention of disease and/or prevention of infection TB vaccine roughly parameterized by the phase 2b trial of M72/AS01E. The calibrated models on which the study is based are published in peer-reviewed literature and referenced appropriately. Overall, the paper is well written, clear in its conclusions and it highlights the need to consider both prevention of infection and prevention of disease vaccines in different epidemiological settings.

Response: We thank the reviewer for their positive feedback.

- 1. An area which would benefit from further clarification are the assumptions around future demographics and non-TB death rates, given the long time horizon of 2050 (26 years of intervention).**

Response:

These are described in detail in the referenced epidemiological model (Harris et al 2020, Science Translational Medicine), on P7/8 of the main paper, and in detail on P21-24 of the appendix. In brief, future population demographics were reproduced by parameterization with UN population division birth and probability of death data and projections for the historical and future periods, respectively.

We have added the text in red to ensure the text in the methods better signposts where detail on the model demographics can be found:

*“A full description of the epidemiological model has been published previously¹⁰, with model structure and parameterisation detailed on page 7 of the previous publication, and **demographics**, calibration and analysis on page 8.”*

We have also added the following to P2 of the manuscript appendix to provide more detail:

“Model inputs to reproduce the age-stratified UN demographic estimates and predictions were the UN population division birth rates per 1000 population and probability of death for an individual of a given age group in a given time period for 1950-2050. A manually adjusted calibration factor for background mortality was employed to reflect migration. “

- 2. As well, as the cost-effectiveness results will be strongly dependent on duration of protection, the paper would benefit from some sensitivity analysis around this.**

Response:

We thank the reviewer for this suggestion. We have revised the manuscript and appendix to now include a sensitivity analysis, exploring the impact on cost-effectiveness results if

duration of protection was 5 or 10 years instead of the 15 years assumed in the base case.

In methods, we have added the following text in bold:

*“Duration of protection was assumed as 15 years waning instantly at the end of the protection in the base case, **though we also explore 5- and 10-year duration of protection in sensitivity analyses.**”*

In results, we have added the following paragraph:

“Duration of protection was not an important determinant of cost-effectiveness in South Africa, only one vaccination scenario became cost-ineffective with a 5 or 10 year duration of protection. In India, a vaccine with five-year duration of protection was not cost-effective in any scenario, whilst vaccinating 10 year-olds became cost-ineffective when a ten year duration of protection was assumed (Appendix figures 12 and 13).”

Figures 12 and 13 in the appendix show the results of this sensitivity analysis.

- 3. It would also benefit from moving discussion of cost-effectiveness thresholds from the supplementary material to the main text.**

Response:

Thank you for this suggestion - we have moved this information to the methods section of the main text (pages 15 and 16) from the appendices.

- 4. In the intervention scenarios a 15 year duration of protection is assumed based on expert opinion. As noted, efficacy trials are 3 years in duration. It would be helpful to expand on any evidence or at least a hypothesis for this assumption (eg. decay rate of T-cell responses etc). As well given the uncertainty here, some additional sensitivity analysis would be informative, such as what is the minimum duration of protection to be cost-effective under the different thresholds? (slightly beyond scope of the paper perhaps) But at what point does this push things to a place where targeting adult pops might be cost-effective but not adolescents (considering realistic coverage levels for groups).**

Response:

We thank the reviewer for this thoughtful reflection on some potential further research questions. Given immune responses to the M72 vaccine are maintained at 3 years post-vaccination, we believe it is a fair assumption that the duration of protection would be longer. Experts in the field indicated that they believed a 15 year duration would be a suitable assumption. As suggested by this reviewer in the previous comment, we have now included a sensitivity analysis for 5 and 10 years duration of protection in the paper to explore alternative assumptions and demonstrate how this would affect the results.

With regards to the suggestion of exploring the minimum duration of protection to be cost effective and the tipping point between targeting adults and adolescents, we like these research questions very much, and will seriously consider these for future research, but they are out of scope of this particular research question and manuscript.

- 5. As the time horizon is 26 years, even considering discounting, changes in future death rates can impact estimates of DALYs averted. A short description of how non-TB death rates are extrapolated to the future would make this transparent. Are non-TB death rates extrapolated linearly in terms of current trends, or with some diminishing returns, or just held constant. What are assumptions around both HIV specific death as well as ART coverage, HIV prevalence. A concise statement of these with a reference to the calibrated model would suffice. It is noted there is discussion of HIV assumptions and 90-90-90 targets in the Discussion section but including this in the methods would be useful.**

Response:

We thank the reviewer for this question and suggested clarification. The historical and future death rates were parameterised using UN population division estimates and projections for probability of death for an individual of a given age group in a given time period for 1950-2050. HIV epidemiology was parameterized using age- and year-specific HIV incidence and AIDS-related mortality for South Africa from Spectrum data (historical period) and projections (future period). (Stover et al 2010; Stover et al 2017) The impact of ART was incorporated by weighting of the relevant TB natural history parameters in the HIV stratum by the ART coverage (details of which parameters can be found on P26 of the appendix of Harris et al 2020, Science Translational Medicine). From 1990-2016, ART coverage in children and adults was parameterised separately based upon data available from Spectrum. (Stover et al 2008; Stover et al 2010) Between 2016 and 2022, ART coverage in both age groups was scaled up to meet the 90% coverage target of the 90:90:90 targets, (SANAC 2017) and was held at 90% from 2022 onwards.

*The background mortality methods have been added to the manuscript as described in response to Q1 by this reviewer. The HIV death rates are described in P12 of the methods and P12 of the appendix of this manuscript, but we have added the text in red to provide additional clarity on HIV epidemiology and ART: “HIV epidemiology was parameterised using age-and year-specific HIV-incidence and AIDS-related mortality **for South Africa from Spectrum data and projections, and ART was parameterised historically using data from Spectrum, scaled up to 90% coverage by 2022 to meet the 90:90:90 targets, and held at 90% coverage beyond 2022**”.*

- 6. Moving some of the section on how CE thresholds are set from supplementary material to the main text would be helpful to the readers to understand the reasoning behind them.**

Response:

Thank you for this suggestion - we have moved this information to the methods section on P14 of the main text from the appendices.

- 7. A 20% relative reduction in efficacy is assumed in HIV positive relative to HIV negative populations. Currently there is no efficacy trial data for this. Does this assumption implicitly take into account levels of viral suppression?. If so, shouldn't this change in time based on assumptions around ART coverage and viral suppression?**

Response:

As the reviewer indicates, there are no efficacy clinical trial data yet available for TB vaccines in HIV positive populations. Therefore, the assumption of 20% reduction in efficacy was based upon clinical trial literature in HIV positive populations from other vaccines. Clinical efficacy of other vaccines has been demonstrated in HIV positive populations, such as influenza and PCV7 vaccines, even at CD4 counts <200. (Crum-Cianfione et al 2014; Nicolini et al 2015) Markers of immunogenicity have indicated a range between no reduction up to 20% reduction in vaccine immunogenicity in HIV positive populations compared to HIV-negative populations (e.g. HPV, Hepatitis A, and Hepatitis B). (Crum-Cianfione et al 2014; Nicolini et al 2015) Therefore, given the lack of data for TB vaccines, the assumption of 20% reduction in efficacy has been taken to ensure our estimates of vaccine impact remain conservative. We have added the following in the limitations section of the paper to ensure this is clear to the reader:

"Few data are available for TB vaccination in HIV-positive populations, so reduction in efficacy in this population was assumed based upon data from other vaccines, which was likely a conservative assumption, leading to a possible underestimate of cost effectiveness in South Africa."

This does somewhat implicitly take into account levels of viral suppression, and may improve over time with increased populations on ART, but given it is a relatively broad and conservative assumption, it is too early to model this more precisely. We will most certainly include this in future work once clinical trial data in HIV-positive populations become available, and have also reflected this with the following additional text in the limitations section:

"Efficacy in HIV positive populations should be explored in future work once data in this population become available."

- 8. It would be more clear in the table if efficacy were split into pre and post and the numbers listed. I take it the sum of efficacies is the same (ie pre + post) across the scenarios for comparison?**

Response:

Thank you, we have clarified this in table 2 by moving the efficacy information from the table notes to the cells, which now list "Efficacy by infection status" more precisely as either "50% pre and post infection" or "50% post infection only".

9. “When patient costs were also included (societal perspective), all scenarios became more cost-effective .. “ The wording in this case could be ambiguous since this is true in the definitional sense, it could read like a result.

Thank you, noted. We have removed the statement that all scenarios became more cost-effective as the reviewer is correct that this is true in the definitional sense.

This paragraph now reads:

“When patient costs were also included (societal perspective), all scenarios became more cost-effective, and the pre- and post-infection vaccine among 15 and 18 year-olds became cost-saving, or dominant, in all scenarios modelled (Table 1, Appendix figure S8), though we note that this is compared to the cost-effectiveness threshold based on health system costs.”

References:

Bunyasi EW, Mulenga H, Luabeya AKK, Shenje J, Mendelsohn SC, et al. (2020) Regional changes in tuberculosis disease burden among adolescents in South Africa (2005–2015). PLOS ONE 15(7): e0235206. <https://doi.org/10.1371/journal.pone.0235206>

N. F. Crum-Cianflone, M. R. Wallace, Vaccination in HIV-infected adults. AIDS Patient Care STDs 28, 397–410 (2014).

Harris RC, Sumner T, Knight GM, Zhang H, White RG. Potential impact of tuberculosis vaccines in China, South Africa, and India. Sci Transl Med. 2020 Oct 7;12(564):eaax4607. doi: 10.1126/scitranslmed.aax4607.

L. A. Nicolini, D. R. Giacobbe, A. Di Biagio, C. Viscoli, Insights on common vaccinations in HIV-infection: Efficacy and safety. J. Prev. Med. Hyg. 56, E28–E32 (2015).

RNTCP, India Annual TB Report 2019.

<https://www.tbcindia.gov.in/WriteReadData/India%20TB%20Report%202019.pdf>

SANAC (2017). “Let our actions count - South Africa’s National Strategic Plan for HIV, TB and STIs 2017-2022.” Retrieved 5th February, 2018, from

http://sanac.org.za/wp-content/uploads/2017/05/NSP_FullDocument_FINAL.pdf.

South African Medical Research Council. First National TB Prevalence survey, South Africa 2018

https://www.knowledgehub.org.za/system/files/elibdownloads/2021-02/A4_SA_TPS%20Short%20Report_10June20_Final_highres.pdf

Statistics South Africa, Education series Volume VII, 2018.

<http://www.statssa.gov.za/publications/92-01-07/92-01-072018.pdf>

J. Stover, P. Johnson, B. Zaba, M. Zwahlen, F. Dabis, R. E. Ekpini, The Spectrum projection package: Improvements in estimating mortality, ART needs, PMTCT impact and uncertainty bounds. Sex. Transm. Infect. 84 (Suppl 1), i24–i30 (2008).

J. Stover, P. Johnson, T. Hallett, M. Marston, R. Becquet, I. M. Timaeus, The Spectrum projection package: Improvements in estimating incidence by age and sex, mother-to-child transmission, HIV progression in children and double orphans. Sex. Transm. Infect. 86 (suppl. 2), ii16–ii21 (2010).

J. Stover, T. Brown, R. Puckett, W. Peerapatanapokin. Updates to the spectrum/ estimations and projections package model for estimating trends and current values for key HIV indicators. AIDS 31 (Suppl 1), S5–S11 (2017).

Uppada DR, Selvam S, Jesuraj N, et al. Incidence of tuberculosis among school-going adolescents in South India. BMC Public Health. 2016;16:641. Published 2016 Jul 26. doi:10.1186/s12889-016-3342-0

REVIEWERS' COMMENTS

Reviewer #2 (Remarks to the Author):

I am satisfied that the reviewers have address all questions and reflected the necessary changes in the manuscript. I recommend publication.

Reviewer #3 (Remarks to the Author):

The authors have thoroughly addressed the points raised in my initial review of the manuscript.